# Cyclopropane-Containing Specialized Metabolites from the Marine Cyanobacterium cf. *Lyngbya* sp.

**DOI:** 10.3390/molecules28093965

**Published:** 2023-05-08

**Authors:** Nurul Farhana Salleh, Jiale Wang, Binu Kundukad, Emmanuel T. Oluwabusola, Delia Xin Yin Goh, Ma Yadanar Phyo, Jasmine Jie Lin Tong, Staffan Kjelleberg, Lik Tong Tan

**Affiliations:** 1Natural Sciences and Science Education, National Institute of Education, Nanyang Technological University, 1 Nanyang Walk, Singapore 637616, Singapore; nie21.nfs@e.ntu.edu.sg (N.F.S.); darylwang@hotmail.com (J.W.); delia.goh.xinyin@gmail.com (D.X.Y.G.); zaara.yadanarphyo@gmail.com (M.Y.P.); tongjielinjasmine@gmail.com (J.J.L.T.); 2Singapore Centre for Environmental Life Sciences Engineering, Nanyang Technological University, 60 Nanyang Drive, Singapore 637551, Singapore; binu.kundukad@ntu.edu.sg (B.K.); laskjelleberg@ntu.edu.sg (S.K.); 3Marine Biodiscovery Centre, Department of Chemistry, University of Aberdeen, Aberdeen AB24 3FX, UK; emmanuel.oluwabusola2@abdn.ac.uk; 4School of Biological Sciences, Nanyang Technological University, 60 Nanyang Drive, Singapore 637551, Singapore; 5School of Biological, Earth and Environmental Sciences, University of New South Wales, Kensington, NSW 2033, Australia

**Keywords:** marine cyanobacterium, cyclopropane-containing metabolites, lyngbyoic acid, *Pseudomonas aeruginosa*, quorum-sensing inhibitor, anti-biofilm

## Abstract

Marine cyanobacteria are known to produce structurally diverse bioactive specialized metabolites during bloom occurrence. These ecologically active allelochemicals confer chemical defense for the microalgae from competing microbes and herbivores. From a collection of a marine cyanobacterium, cf. *Lyngbya* sp., a small quantity of a new cyclopropane-containing molecule, benderadiene (**2**), and lyngbyoic acid (**1**) were purified and characterized using spectroscopic methods. Using live reporter quorum-sensing (QS) inhibitory assays, based on *P. aeruginosa* PAO1 *lasB-gfp* and *rhlA-gfp* strains, both compounds were found to inhibit QS-regulated gene expression in a dose-dependent manner. In addition to lyngbyoic acid being more active in the PAO1 *lasB-gfp* biosensor strain (IC_50_ of 20.4 µM), it displayed anti-biofilm activity when incubated with wild-type *P. aeruginosa*. The discovery of lyngbyoic acid in relatively high amounts provided insights into its ecological significance as a defensive allelochemical in targeting competing microbes through interference with their QS systems and starting material to produce other related analogs. Similar strategies could be adopted by other marine cyanobacterial strains where the high production of other lipid acids has been reported. Preliminary evidence is provided from the virtual molecular docking of these cyanobacterial free acids at the ligand-binding site of the *P. aeruginosa* LasR transcriptional protein.

## 1. Introduction

Cyanobacteria are a group of Gram-negative photoautotrophic prokaryotic organisms that can photosynthesize and are known botanically as Cyanophyta or ‘blue-green algae’. These microbes hold a fossil record of 3.3 to 3.5 billion years and are found in abundance in diverse ecosystems [1]. Filamentous marine cyanobacteria are an important source of lead compounds for drug discovery and development, particularly as anticancer therapeutics [2]. This is due to the cyanobacterial compound’s ability to target specific eukaryotic cytoskeletons, which can modulate cell apoptosis and cell death in cancerous cells. Some of these bioactive secondary metabolites have also been reported to target biological enzymes, such as histone deacetylases, which affect the regulation of transcription [2]. For instance, the potent cytotoxic marine cyanobacterial compound dolastatin 10 has been used as a template for the synthesis of monomethyl auristatin E (MMAE) [3]. MMAE was then incorporated as a warhead in the synthesis of the antibody–drug conjugate (ADC) brentuximab vedotin, used for the treatment of Hodgkin’s lymphoma and systemic anaplastic large-cell lymphoma [4]. In addition to brentuximab vedotin, at least four other ADCs based on the MMAE warhead are currently approved for cancer treatment (https://www.marinepharmacology.org/approved/, accessed on 13 February 2023). In recent years, bioactive specialized metabolites from filamentous marine cyanobacteria have been found to possess significant anti-infective properties, such as antimalarials and antibacterial activities [5]. An example is gallinamide A, which was found to be a potent cathepsin L inhibitor and is currently being evaluated, along with its analogs, as potential anti-trypanosomatid and antiviral drugs [6,7].

Cyclopropane-containing specialized molecules have been reported from marine cyanobacteria, although their occurrence is uncommon. Of the almost 1000 marine cyanobacterial compounds, only 1.3% (13 compounds) contain a cyclopropane unit. Examples of marine cyanobacterial cyclopropane-containing metabolites include curacins A–D, grenadadiene, debromogrenadadiene, grenadamide, hoshinolactam, cocosolide, coibacins A and B and free acid forms, e.g., lyngbyoic acid and majusculoic acid [8,9,10,11,12,13,14,15,16,17]. Interestingly, an analog of the curacin class of compounds, curacin E, was reported from the brittle star, *Ophiocoma scolopendrina*, suggesting that its origin could be derived from symbiotic microbes [18]. These cyanobacterial compounds are produced as allelochemicals in response to competing organisms and, hence, possess a range of biological activities. For instance, potent cytotoxic activities have been reported for the curacin series of molecules [8,9,10] and grenadadiene [11], while anti-microbial properties are associated with Lyngbyoic acid [17] and majusculoic acid [19]. In addition, coibacins and hoshinolactam have significant antiprotozoal properties, while the symmetrical glycosylated macrolide dimer cocosolide possesses immunosuppressant activity.

Apart from its occurrence in marine cyanobacteria, the strained cyclopropane subunits are found in many complex natural products, and such a structural motif confers diverse biological activities, including cytotoxicity, anti-infectives and immunosuppressive properties [20,21]. Moreover, the cyclopropyl moiety is a common structural feature present in preclinical/clinical drug compounds. This is due to several beneficial features attributed to the cyclopropyl ring, including the coplanarity of the three carbon atoms, the enhanced pi-character of C-C bonds and having shorter and stronger C-H bonds compared with C-H bonds in alkanes [22,23]. The cyclopropane moiety has also been reported to provide special steric, electronic and stereo-electronic effects, as well as being able to serve as the moiety responsible for some biological processes [24]. Moreover, the fused cyclopropane unit is functionally relevant and confers resistance to β-oxidation. Because of the significance of the cyclopropane unit, it is not surprising that it is the 10^th^ most frequently found ring system in small molecule-based drugs [22].

Given the biomedical potential of marine filamentous cyanobacteria, this study reports on the isolation of rare cyclopropane-containing specialized metabolites, including lyngbyoic acid (**1**) and a new compound, benderadiene (**2**). In addition, these compounds were evaluated for their anti-quorum-sensing activity based on two *Pseudomonas aeruginosa* PAO1 biosensor strains, namely, *lasB-gfp* and *rhlA-gfp*, as well as the anti-biofilm activity of lyngbyoic acid. The ecological significance of their discovery, as well as related modified fatty acids previously reported from other marine cyanobacterial strains, is also discussed.

## 2. Results and Discussion

### 2.1. Extraction, Fractionation, Bioassay Evaluation and Preliminary Dereplication of Marine Cyanobacterial Specialized Metabolites

An occurrence of a filamentous marine cyanobacterial bloom was observed on 17 October 2019 in the vicinity of Tanjong Hakim, St. John’s Island, Singapore. Samples of the microalga were subsequently collected and stored in MeOH at −20 °C before a chemical workup. Morphological observation under a light microscope showed that the filamentous marine cyanobacterium could be that of *Lyngbya majuscula* because of its long, filamentous form and colorless sheath with brownish-green trichomes. An initial 1 L of the marine cyanobacterial sample was extracted exhaustively three times, each extraction performed using CH_2_Cl_2_/MeOH (2:1) with gentle heating at 50 °C. The combined extracts were dried in vacuo and subjected to normal-phase vacuum liquid chromatography (NP-VLC) to obtain eight fractions using a combination of eluting solvent hexanes, EtOAc and MeOH in increasing polarity. These VLC-derived fractions were then subjected to a brine shrimp (*Artemia salina*) toxicity assay tested at 1000 ppm and 100 ppm. It was observed that mid-polarity VLC-derived fractions 4 and 5 had significant toxicity at 98% and 100% lethality, respectively, when tested at 100 ppm. In addition, the VLC fractions were evaluated for anti-quorum-sensing activity based on the *Pseudomonas aeruginosa* PAO1 *lasB-gfp* reporter biosensor strain. It was found that fraction 3 had the highest inhibition at 100 µg/mL, followed by fractions 4 and 5 with similar fluorescence inhibitory activity (Appendix A).

The metabolomics approach, the study of all small molecules produced from the cellular metabolic functions of a living organism, tissues or cells, has been utilized in the study of natural products [25]. In particular, the use of tandem mass spectrometry (MS/MS) is an important tool in dereplication processes for natural product research [26]. Understanding MS/MS fragmentation through the visualization of molecular networks allows for efficient compound dereplication as well as the detection of potential new molecules present in the sample of interest. In this study, compound dereplication of cyanobacterial fractions was performed using an MS-based molecular network generated on Cytoscape (Figure 1). Based on the MS-molecular networking on GNPS, a total of 2321 nodes were detected from 8 VLC-derived fractions, and several identified molecular families were dereplicated based on MS spectral matches with the GNPS database and MarinLit (Figure 1). These known cyanobacterial compounds include chlorophyll a, pheophytin a, saliniketal A and pukeleimides, a class of 5-ylidenepyrrol-2(5H)-ones [27]. Additional compound dereplication was performed on selected QSI-active VLC fractions, 3 and 4, using an Agilent 1200 series HPLC-UV system (Agilent, Santa Clara, CA) coupled with ESI-TOF-MS (Appendix A). Based on manual dereplication using MarinLit, five known hits corresponded to the cyanobacteria-derived compounds isolated from *Lyngbya majuscula* (Appendix A). Aside from the LC-MS detection and isolation of the metabolites lyngbyoic acid (**1**) and benderadiene (**2**), manual dereplication analysis led to an additional 10 unknown compounds identified mainly from the base peak chromatograms of VLC fractions 3 and 4 (Appendix A). For instance, a molecular ion at *m/z* 429.1603 [M+H]^+^ revealed an isotopic cluster, suggesting the presence of one bromine atom with a molecular formula of C_21_H_34_BrO_4_ [M+H]^+^, which was detected in VLC fraction 3 (Appendix A). This molecular mass ion is considered a new derivative of the cyclopropyl-containing compound grenadadiene, based on a characteristic fragmentation pattern that showed a fragment ion at *m/z* 195.1766 (C_13_H_23_O) and a mass difference of 14 Da, denoting an additional ethylene group in its fatty acid chain [11]. In addition, three hydrophobic unknown molecules with molecular ions at *m/z* 609.2685, 637.3001 and 607.2916 in both VLC fractions 3 and 4 were detected. Considering the HRESI ammonia adducts observed at *m/z* 631.2502, 659.2811 and 629.2721 for the three compounds, the molecular formulae C_35_H_37_N_4_O_6_, C_37_H_41_N_4_O_6_ and C_36_H_39_N_4_O_6_ were assigned, respectively. These molecules are putatively new derivatives of the pheophorbide-like compound, which was also detected with a molecular ion at *m/z* 623.2858 [M+H]^+^ (C_36_H_39_N_4_O_6_), with an observed intense base peak in fractions 3 and 4 (Appendix A). Interestingly, a similar pheophorbide-like compound, with anti-herpes simplex viral activity, was previously isolated from the marine green alga *Dunaliella primolecta* [28]. As there were several unknown clusters in the compound dereplication analysis, we proceeded to pursue these fractions for the isolation of possible new specialized metabolites.

### 2.2. Isolation and Structural Elucidation of Cyclopropane-Containing Specialized Metabolites

Based on the bioassay data, along with preliminary ^1^H NMR analysis of the bioactive VLC fractions, fraction 5 (eluted with 60% EtOAc/Hex) was selected for the isolation of bioactive molecules. Proton NMR analysis of this fraction revealed high-field signals below 1 ppm, attributed to cyclopropane-associated protons. Fraction 5 was separated using reversed-phase solid-phase extraction with 90% MeOH to remove pigments. Reverse-phased C-18 HPLC was then conducted on the filtered fraction, and compound **2** was collected at a retention time of 25 min, yielding about 0.5 mg. The additional extraction of about 500 mL of the filamentous marine cyanobacterial sample was made from the remaining sample available, and the fraction selected for further purification was guided by ^1^H NMR, which led to the isolation of about 39.4 mg of lyngbyoic acid (**1**). The structure of isolated lyngbyoic acid was confirmed via comparison with the published NMR data [17].

Based on HR-MS data, **2** provided a [M+H]^+^ protonated molecule at *m/z* 357.1422, which is consistent with the molecular formula C_18_H_29_BrO_2_, requiring four degrees of unsaturation. Based on the preliminary analysis of 1D NMR spectra, there is the presence of olefinic protons at δ 5.67, δ 5.84, δ 6.16 and δ 6.27 in the ^1^H NMR spectral data (Table 1). Along with these peaks, the high-field proton signals associated with the cyclopropane moiety at δ 0.43 and δ 0.20 were also observed, similarly seen in grenadadiene [11]. The presence of ^13^C NMR signal peaks at δ 128.7 and δ 173.2 potentially indicated the presence of a bromine atom linked to an olefinic carbon atom and carbonyl carbon bonded to oxygen, respectively (Table 1).

From the 2D NMR data, including ^1^H-^1^H COSY and ^1^H-^13^C HMBC, partial structures **2a** and **2b** were generated (Figure 2). Partial structure **2a** was determined similarly to grenadadiene, in which we observed that the partial molecular mass of about *m/z* 195 correlated to the cyclopropyl fatty-acid moiety C_13_H_23_O. The ^1^H NMR spectral data of this moiety is similar to grenadadiene as well. Partial structure **2b** was elucidated through ^1^H-^1^H COSY and HMBC correlations. The germinal protons at δ 4.70 (H-1’) were found to have ^1^H-^1^H COSY correlations with a neighboring olefinic proton resonating at δ 6.16 (H-2’) as a doublet of a triplet peak, indicating the correlation with another olefinic proton at δ 6.27 (H-3’), which resonated as a doublet. This doublet indicated no further correlation with another proton but had a ^1^H-^13^C HMBC correlation with the olefinic carbon at δ 128.7 (C-4’), which is proof of an attachment to a bromine atom. Furthermore, there is no indication of any proton attached to quaternary C-4’ due to the absence of ^1^H-^13^C HSQC correlation. Lastly, a ^1^H-^13^C HMBC correlation was found between two terminal olefinic protons, namely, δ 5.67 and δ 5.84 (H_2_-5’) to δ 128.7 (C-4’).

The H-5’ protons resonating as two broad singlets with ^1^H peaks at δ 5.67 and δ 5.84 were supported by the coupling between these two protons due to the presence a of neighboring bromine atom attached to C-4’ at δ 128.7. Furthermore, based on the ^1^H-^13^C HSQC correlation, the terminal olefinic carbon signal at δ_C_ 121.0 (C-5’) was found to be attached to these two olefinic protons. To further support this analysis, based on a search in MarinLit, a similar chemical shift pattern was observed in the marine natural product 7-bromo-myrcene, isolated by Ichikawa and co-workers in 1974, in which the two protons δ 5.59 and δ 5.85 attached to the terminal sp^2^ carbon atom with bromine attached to the adjacent olefinic carbon, as seen in Figure 2 [29]. These analyses, therefore, elucidated the partial structure of **2b**.

A coupling constant of 16.0 Hz was found between H-2’ and H-3’, which supports a trans-geometry structure of the olefinic bond. This contrasted with the olefinic protons in grenadadiene, which had a coupling constant of 6.5 Hz to support a *cis*-geometry for the compound. Furthermore, based on NOESY data, it was also found that there was no correlation between H-2’ and H-3’, which further supports the *trans*-geometry of the structure. The stereochemistry of the cyclopropane moiety in **2** is proposed to be identical to that of grenadadiene and is supported by the isolation of free acid **1** having a similar optical rotation value to the published data for lyngbyoic acid by Kwan et al. [17]. However, due to an insufficient amount of **2**, we were not able to confirm the absolute stereochemistry of the cyclopropane unit, which would involve the acid hydrolysis of the ester bond, the isolation of the free acid and the determination of its optical rotation value. As such, the relative stereochemistry of 4*R*,6*R* in **2** is depicted instead. With the elucidation of the partial structures of this compound, along with the HMBC correlation of the H_2_-1’ protons at δ 4.70 to C-1 at δ 173.2, the structure of **2** is, hence, proposed as depicted in Figure 3 and given the trivial name of benderadiene [= (*E*)-4-bromopenta-2,4-dienyl 3-((1*R**,2*R**)-2-heptylcyclopropyl)propanoate].

### 2.3. Biological Activity of Cyclopropane-Containing Metabolites

The anti-QS activities of cyclopropane-containing compounds **1** and **2** were evaluated for their ability to inhibit QS-controlled green fluorescent protein (GFP) expression using two biosensor strains of *Pseudomonas aeruginosa* PAO1, where either the *lasB* or *rhlA* promoter was fused to an unstable *gfp* (ASV). In these reporter strains, the production of GFP is indicative of QS induction. QS inhibitory activity in the compounds is then reflected in a reduction in GFP production relative to the control. The GFP expression was measured in relative fluorescence units and normalized by dividing the GFP values by the corresponding OD_600_ value measured at that time point. Compounds **1** and **2** were tested in a dose-dependent manner via incubation with either *P. aeruginosa* PAO1 *lasB-gfp* or the *rhlA-gfp* biosensor strain. Compound **1** was found to have inhibitory activity with an IC_50_ of 20.4 µM when tested in PAO1 *lasB-gfp* but no observable inhibitory activity with PAO1 *rhlA-gfp*. The quorum-sensing inhibitory effect of lyngbyoic acid in this study is consistent with the findings of Kwan and co-workers, where **1** was found to strongly affect LasR when screened against a panel of reporters based on four different AHL receptors, including LuxR, AhyR, TraR and LasR [17]. Benderadiene (**2**), however, exhibited reduced inhibitory activities with IC_50_ values of 89.9 µM and 80.3 µM in PAO1 *lasB-gfp* and PAO1 *rhlA-gfp*, respectively. In addition, **1** was tested in a brine shrimp toxicity assay and was found to have an LC_50_ of 18.0 mM. Unfortunately, the brine shrimp toxicity of benderadiene was not evaluated due to insufficient amounts.

Lyngbyoic acid (**1**) was previously shown to reduce the pyocyanin and elastase production and transcription levels in wild-type *P. aeruginosa* [17]. In addition, the global transcriptional effects of lyngbyoic acid somewhat replicate the gene expression changes of *P. aeruginosa* during chronic lung infections in cystic fibrosis patients. For the effects of **1** on biofilm genes, they were found to downregulate certain members of the *psl* operon, such as *pslA* and *pslB*, as well as *pslN*, while it upregulated the entire *pelABCDEFG* operon, which is needed for the synthesis of a glucose-rich matrix exopolysaccharide, an important component of bacterial biofilms [30]. In this study, lyngbyoic acid was tested against wild-type *P. aeruginosa* biofilm formation. *P. aeruginosa* biofilms were grown for 24 h in the presence of different concentrations of **1** (Figure 4). Biofilms were imaged using a confocal microscope, and the total biovolume (green and red channels) was quantified (Figure 5). A significant reduction in biovolumes was observed when they were treated with concentrations above 500 µM of **1**. However, there seems to be no effect from **1** on preformed biofilms.

### 2.4. Molecular Docking of Marine Cyanobacterial Cyclopropane-Containing Metabolites

Molecular docking has been used as a tool to visualize the interaction between a ligand and a target protein of interest [31]. Since some degree of quorum-sensing inhibitory activities were observed for compounds **1** and **2**, these molecules could be hypothesized to have some binding affinity for LasR, a transcriptional activator of the *P. aeruginosa* quorum-sensing system. Preliminary molecular docking on Swissdock was, therefore, conducted using the X-ray structure of the *P. aeruginosa* LasR ligand-binding domain (PDB ID 2UV0), and it was found that **1** and **2** deposited within the ligand-binding domain of LasR in a similar way to the native autoinducer *N*-3-oxo-dodecanoyl-L-homoserine lactone (Figure 6). Moreover, this is supported by a study by Kwan et al., where lyngbyoic acid was found to inhibit the response of LasR-based QS reporter plasmids to the natural ligand *N*-3-oxo-dodecanoyl-L-homoserine lactone [17]. We also considered comparing other natural analogs of **2**, such as grenadadiene, debromogrenadadiene and grenadamide, for the molecular docking experiments, but these related compounds were not found to bind at the ligand-binding site of the LasR protein. The relatively larger molecular sizes of grenadadiene and debromogranadadiene or the presence of a benzene ring in granadamide could have prevented these molecules from binding at the ligand-binding site of LasR. Interestingly, the cyclopropane-containing modified fatty acid majusculoic acid was found to be docking at the LasR-ligand binding domain (Appendix A). This could suggest a higher probability of molecules docking at the ligand binding site of LasR if the cyclopropane-containing compounds are either in the free acid forms or of a certain molecular size.

### 2.5. Ecological Significance of Marine Cyanobacterial Lyngbyoic Acid and Other Modified Fatty Acids

The significant amount of lyngbyoic acid obtained from a bloom-occurring marine cyanobacterial sample in this study sheds light on the ecological importance of this lipid acid. In a study by Kwan and co-workers, the yield of lyngbyoic acid was reported to be high at 1.32% of the lipophilic fraction prepared from *L.* cf. *majuscula* [17]. This lipid acid was also revealed to strongly affect the AHL receptor LasR and reduce pyocyanin and elastase (LasB) in both the protein and transcript levels in wild-type *P. aeruginosa*. It has been suggested that large amounts of lyngbyoic acid are required in the native cyanobacteria since natural quorum-sensing inhibitors, such as honaucins, malyngolide and tumonoic acids, usually show IC_50_s in the micromolar range [32,33,34]. This observation is consistent with this study, where lyngbyoic acid was found to have an IC_50_ of 20.4 µM when tested on the PAO1 *lasB-gfp* biosensor strain. In addition, the high concentration of lyngbyoic acid could overcome the rate-limiting step by serving as a ready substrate for the biosynthesis of a range of related metabolites with diverse ecological functions. It was found that acyl–acyl carrier protein synthetase is involved in the activation and recycling of exogenous fatty acids in certain cyanobacterial strains, such as *Synechocystis* sp. PCC 6803 and *Synechococcus elongatus* PCC 7942 [35]. Similar enzymatic systems could be involved in the recycling of modified fatty acids for the production of analogs. To date, only four other cyclopropane-containing compounds, i.e., grenadadiene, debromogrenadadiene, granadamide and benderadiene, have been reported with a lyngbyoic acid tail. The actual number is probably higher, as analysis of the molecular networking of VLC-derived fractions in this study detected several related precursor ions in lower quantities, which were found within the lyngbyoic acid and benderadiene molecular families (Figure 7).

The ecological importance of lyngbyoic acid in benthic marine cyanobacteria has been implicated in at least two studies. Firstly, research by Sneed and co-workers reported the isolation of lyngbyoic acid along with other specialized molecules, including malyngolide, microcolins A–B and desacetylmicrocolin B, from an assemblage of four unique cyanobacterial species collected during a bloom season at Indian River Lagoon, Florida [36]. As these molecules were found to be active in ecologically relevant assays, their chemical defenses contribute to the persistence of microalgal blooms in the lagoon during the summer seasons. For instance, lyngbyoic acid strongly reduced the growth of two marine fungal strains, *Lindra thalassiae* and *Dendryphiella salina*, at a natural concentration of 0.73 mg mL^−1^ and inhibited urchin feeding at 5.69 mg g drwt^−1^ [36]. Another study by Engene et al. suggested that the persistent presence of antibacterial compounds, including lyngbyoic acid, lyngbic acid and malyngolide, could be an ecological adaptation of members belonging to the recently described marine cyanobacterial genus *Dapis* [37]. The almost completely clean surfaces of several *Dapis* species, revealed by SEM images, could be due to the production of these specialized compounds. Interestingly, a similar chemical profile was observed in this study where both the C-14 lyngbic acid and malyngolide were detected in the compound dereplication analysis of the selected VLC fractions (Appendix A).

Similar defensive strategies could be used by other marine cyanobacterial strains in producing high amounts of unique modified fatty acids. These modified fatty acids include lyngbic acids (**3**–**5**) [38,39,40], malyngic acid (**6**) [41], (2*R*)-2,5-dimethyldodecanoic acid (**7**) [42], pitinoic acid (**8**) [43], dysidazirine carboxylic acid (**9**) [44], puna’auic acid (**10**) [45], 11-oxopalmitelaidic (**11**) [46], 2-methyldecanoic acid and 2-methyldodecanoic acid [47] (Figure 8 and Table 2).

In particular, lyngbic acids, e.g., 7-methoxydodec-4(*E*)-enoic acid (**3**), 7(*S*)-methoxytetradec-4(*E*)-enoic acid (**4**) and 7-methoxy-9-methylhexadeca-4(*E*),8(*E*)-dienoic acid (**5**), are used to form the malyngamide class of compounds [48,49]. There are two distinct portions present in the malyngamides: a methoxy fatty acid tail and a variety of functionalized amines linked through an amide bond. These lyngbic acids have varying chain lengths, ranging from C-12 to C-20, with a methoxy group at C-7 as well as a trans double bond at C-4. A majority of the reported malyngamides contained the C-14 lyngbic acid chain. The C-16 analog is present in malyngamides D and E, while the C-12 and C-20 analogs are present in malyngamides (e.g., malyngamides G, S, U-W and Y) and serinol-derived malyngamides, respectively. Only the C-12 (**3**), C-14 (**4**) and C-16 (**5**) lyngbic acids have been isolated in free acid forms from marine cyanobacteria [38,39,40]. In addition, these lipid acids could potentially be precursors to cyclized forms, such as the formation of malyngolide and malyngolide dimer from 2,5-dimethyldodecanoic acid (**7**).

A recent study by Moss and co-workers documented the discovery of the type A malyngamide biosynthetic pathway in the sequenced genome of the marine cyanobacterial genus *Okeania* [49]. Most of the type A malyngamides contained a highly decorated, six-membered cyclohexanone head group linked via an amide bond to a methoxylated C-14 lyngbic acid tail. Based on feeding experiments with a ^13^C-labeled substrate, they provided evidence of an intact octanoate unit being transferred to the first ketosynthase module. It could be envisioned that the biosynthesis of lyngbyoic acid could take place in a similar way, initiated by the loading of the intact octanoate unit. The subsequent chain extension by two units of acetyl CoA, the partial reduction of the carbonyl group to a double bond at C-4 and the formation of the cyclopropane unit involving an exogenous C1 unit from SAM via polar or radical chemistry over the double bond of the mono-unsaturated precursor lipid could form lyngbyoic acid [21].

Since lyngbyoic acid has quorum-sensing inhibitory activity, we speculated that other modified fatty acid types could possess similar activity. We, therefore, carried out the molecular docking of selected fatty acids with LasR transcriptional protein and found the docking of these molecules at the ligand-binding site of the protein (Figure 9). The modified lipid acids may provide a chemical defense to the marine cyanobacteria producer by interfering with the quorum-sensing systems of competing microbes. Biological evaluations of actual compounds for quorum-sensing inhibitory properties would need to be carried out to test this hypothesis. Interestingly, both the C-14 lyngbic acid 7(*S*)-methoxytetradec-4(*E*)-enoic acid (**4**) and pitinoic acid (**8**) have been reported to have anti-quorum-sensing activity [43,50]. In addition, the allenic acid puna’auic acid, 11-oxopalmitelaidic and 2-methyldecanoic acid have been reported to be major metabolites of the metabolome of the marine cyanobacteria *Pseudanabaena* sp., *Leibleinia gracilis* and *Trichodesmium erythraeum*, respectively, and are hypothesized to have anti-quorum-sensing or antibacterial activities [45,46,47]. Furthermore, several unsaturated fatty acids, such as palmitic acid, oleic acid, palmitoleic acid and a-linolenic acid, from certain cyanobacterial strains are responsible for significant quorum-sensing inhibitory and anti-biofilm activity against important bacterial pathogens [51,52]. Taken together, the production of unique lipid acids by marine cyanobacteria is probably widespread due to their versatility as defensive/signaling molecules during bloom occurrence as well as a structural template for the synthesis of structurally diverse allelochemicals. It would also be interesting to assess these modified fatty acids as potential biomarkers by relating their production amounts with the dynamics of cyanobacterial blooms.

**Table 2 molecules-28-03965-t002:** Modified lipid acids reported from marine cyanobacteria and their reported amounts, biological activities and number of analogs containing lipid acid tails.

Marine Cyanobacterial Modified Lipid Acid	Organism/Highest Reported Amount	Biological Activity of Lipid Acid/ # of Analogs Containing a Lipid Acid Tail
*Lyngbic acids* 7-Methoxydodec-4(*E*)-enoic acid (**3**) 7(*S*)-Methoxytetradec-4(*E*)-enoic acid (**4**) 7-Methoxy-9-methylhexadeca-4(*E*),8(*E*)-dienoic acid (**5**)	*Lyngbya majuscula*/75 mg *L. majuscula*; *Okeania hirsute*; *Moorea producens*; cyanobacterial mat from black-band consortium/>200 mg in a report by Soares and co-workers [53] *L. majuscula*/10 mg	Not tested/6 Interfered quorum sensing in the *Vibrio harveyi* QS reporters and luminescence in native coral *Vibrio* spp./32 Not tested/2
Malyngic acid (**6**)	*L. majuscula*/766 mg from two collections	Not tested/0
(2*R*)-2,5-Dimethyldodecanoic acid (**7**)	*L. aestuarii*/54 mg	Strongly inhibited the growth of the common duckweed *Lemna minor*/1
Pitinoic acid A (**8**)	*Lyngbya* sp./0.3% of total dry weight	Interfered quorum sensing in *P. aeruginosa* by reducing the transcript levels of *lasB* and the pyocyanin biosynthetic member *phzG1*/1
Lyngbyoic acid (**1**)	*L.* cf. *majuscula*/42.4 mg	Strongly affected the AHL receptor LasR and reduces pyocyanin and elastase (LasB), both on the protein and transcript level in wild-type *P. aeruginosa*; inhibited fungal growth and herbivore feeding/5
Dysidazirine carboxylic acid (**9**)	*Caldora* sp./3 mg	Anti-inflammatory/0
Puna’auic acid (**10**)	*Pseudanabaena* sp./3.8 mg	Not tested/0
11-Oxopalmitelaidic acid (**11**)	*Leibleinia gracilis*/3.5 mg	Not tested/0
2-Methyldecanoic acid (**12**) and 2-methyldodecanoic acid (**13**)	*Trichodesmium erythraeum*/comprising up to 75% of the total fatty acid pool	Not tested/0

## 3. Materials and Methods

### 3.1. General Experimental Procedures

All NMR spectra were recorded in CDCl3 on a 400 MHz Bruker NMR Spectrometer (400.13 MHz 1H, 100.61 MHz 13C) using residual solvent signals as internal references (referenced to residual CDCl3 observed at δH 7.24 or δC 77.0) with chemical shifts given in ppm downfield from TMS. Optical rotations were measured on an Anton Paar Polarimeter while UV absorbance was measured on a PerkinElmer UV-Visible spectrophotometer. The isolation and purification of compounds **1** and **2** were conducted on Shimadzu LC-8A preparative LC coupled to a Shimadzu SPD-M10A VP diode array detector HPLC.

### 3.2. Sample Collection

Marine cyanobacterial samples with cell morphology resembling that of *Lyngbya majuscula* Harvey ex Gomont 1892 (Oscillatoriaceae) were collected by hand during a bloom occurrence of the microalga in October 2019 located at intertidal shores in the vicinity of Tanjong Hakim, St. John’s Island, Singapore (1°13′24.5″ N; 103°50′42.8″ E). Samples were subsequently stored in 70% EtOH at −20 °C before a chemical workup at NIE. The voucher specimen, TLT/SJI/17OCT2019/001, is deposited at Natural Sciences and Science Education, National Institute of Education, Singapore.

### 3.3. Extraction and Isolation of Compounds

An initial marine cyanobacterial sample (ca. 1.0 L, wet weight) was thawed and extracted exhaustively with 2:1 CH_2_Cl_2_/MeOH. After the solvent was evaporated in vacuo, about 3 g of a crude organic extract was obtained. The dried extract was reconstituted in small amounts of MeOH, and sufficient celite was added to make a slurry. The slurry was evaporated in vacuo to powder form for fractionation using vacuum flash chromatography (sintered glass Buchner funnel, 500 mL, diameter 95 mm, height 196 mm) on normal phase Si gel using a stepwise gradient with increasing polarity of 100% hexanes: 9:1 hexanes/EtOAc, 4:1 hexanes/EtOAc, 3:2 hexanes/EtOAc, 2:3 hexanes/EtOAc, 1:4 hexanes/EtOAc, 100% EtOAc, 9:1 EtOAc/MeOH, and 8:2 EtOAc/MeOH. The selected brine shrimp toxic fraction 5, was filtered on a Sep-Pak C18 cartridge (Strata 5 g, Phenomenex, Torrance, CA, USA) using 90% aqueous MeOH to remove pigments. The resulting filtrate was further subjected to semipreparative RP-HPLC separation (Shim-Pack GIST Shimadzu 5mm C-18, 250 × 10 mm, 97.5% acetonitrile in H_2_O at 3.0 mL/min, detected at 210 nm, 230 nm and 290 nm) to yield 0.5 mg (*t*_R_ = 24.9 min) of benderadiene (**2**) as a white amorphous solid. A second marine cyanobacterial sample (ca. 500 mL, wet weight) was subjected to similar extraction and fractionation procedures. Fractions 3 and 4, eluted with 3:2 hexanes/EtOAc and 2:3 hexanes/EtOAc, respectively, were combined and filtered on a Sep-Pak C18 cartridge (Strata 5 g, Phenomenex, Torrance, CA, USA) using 90% aqueous MeOH to remove pigments. The filtered fractions were subjected to semipreparative RP-HPLC separation (Shim-Pack GIST Shimadzu 5 mm C-18, 250 × 10 mm, 97.5% acetonitrile in H_2_O at 3.0 mL/min) to yield 39.4 mg (*t*_R_ = 12.8 min) of lyngbyoic acid (**1**).

### 3.4. Compound Characterization Data

Benderadiene (**2**): white amorphous solid; ${\left[\alpha \right]}_{D}^{20}$  + 484.7 (c 0.02, MeOH); UV (MeOH) λ_max_ (log ε) 225 nm (3.95); ^1^H and ^13^C NMR spectral data (CDCl_3_, 400.13 and 100.61 MHz, respectively), Table 1 and Appendix A; HR-QTOF MS *m/z* 357.1422 [M + H]^+^ (calculated for C_18_H_30_O_2_^79^Br, 357.1429).

### 3.5. Quorum-Sensing Inhibitory Assay

The anti-quorum-sensing bioassay was carried out using *Pseudomonas aeruginosa* PAO1 *lasB-gfp* and *rhlA-gfp* reporter strains. Compounds **1** and **2** were prepared in a 96-well microtiter plate at 10 mM stock concentration dissolved in 100% DMSO, conducted in triplicate. Compounds **1** and **2** were then mixed with ABTGC medium and serial diluted to obtain a concentration of 20 μM in the first dilution factor (with 0.2% of DMSO). A total of seven dilution factors, down to 0.3125 μM, were performed. An overnight culture of PAO1 *lasB-gfp* strain [54], grown in lysogeny broth at 37 °C, 200 rpm, was diluted in ABTGC medium to an optical density of 0.02 at OD_600_, which corresponded to 2.5 × 107 CFU/mL. An equal amount of bacterial suspension was added to reach final test concentrations of 10, 5, 2.5, 1.25, 0.625, 0.3125 and 0.1563 μM. A DMSO control, media control and culture control were used, and the microtiter plates were incubated at 37 °C in a Tecan Infinite 200 Pro plate reader to measure the cell density (OD_600_) and green fluorescence protein fluorescence (excitation at 483 nm, emission at 535 nm) with 15 min intervals for up to 16 h. A similar procedure was carried out using the PAO1 *rhlA-gfp* biosensor strain [55].

### 3.6. Anti-Biofilm Assay

Overnight cultures of GF-labeled *P. aeruginosa* (PAO1) were grown in Luria–Bertani broth (10 g/L NaCl, 10 g/L yeast extract and 10 g/L tryptone) at 37 °C under shaking conditions at 200 rpm. The overnight culture was diluted to an optical density of 0.4 at 600 nm (OD_600_) and added to a glass-bottomed 24-well plate with different concentrations (1–100 µM) of lyngbyoic acid (**1**). The plates were then incubated at 37 °C for 24 h for biofilm formation. The dead bacteria and eDNA in the biofilm were stained with 3 µM propidium iodide (PI; Thermo Fisher Scientific, Waltham, MA, USA) for 15 min. Three-dimensional image stacks of the surface-attached biofilm were acquired using a Carl Zeiss LSM 780 laser scanning confocal microscope with a 20× objective. Further image processing and analysis were conducted using Imaris 9.0 (Bitplane, South Windsor, CT, USA).

### 3.7. Mass Spectrometric-Based Molecular Network of Marine Cyanobacterial VLC-Derived Fractions

VLC-derived marine cyanobacterial fractions were filtered over C18 SPE cartridges by application of a 1.0 mL sample (1 mg/mL) and elution with 3 mL CH_3_CN. Solvent was removed in vacuo using a rotary evaporator before being redissolved in 1 mL CH_3_CN, vortex mixed for 5 min and transferred into separate Eppendorf tubes. Tubes were then centrifuged at 10,000 rpm at 4 °C for 10 min, and the supernatant was aliquoted and diluted with CH_3_CN to 10,000 × dilution. One-and-a-half microliters of each diluted sample was subjected to LC-HRMS/MS (Q Exactive Plus Hybrid Quadrupole-Orbitrap Mass Spectrometer (Thermo Fisher Scientific) equipped with a heated electrospray ionization (H-ESI) probe) performed with a Thermo Scientific Hypersil GOLD (C18 50 mm × 2.1 mm, 1.9 mm) column and maintained at a column temperature of 40 °C and a sample temperature of 4 °C using a gradient elution program of 0.1% aq. HCOOH (mobile phase A) and 98% CH_3_CN in 0.1% aq. HCOOH (mobile phase B) at a flow rate of 0.5 mL/min. The gradient program began at 10% and increased to 50% of mobile phase B within 2 min and was held at 50% of mobile phase B for 2 min. It was then increased to 100% of mobile phase B within 6 min and was held at 100% of B for 0.5 min before reconditioning back to the starting composition in 0.5 min, and it was held at the starting composition of 10% of B for 3 min, bringing the total runtime to 14 min. All mass spectra were collected in the positive ion and data-dependent acquisition mode, where the first five most intense ions of each full-scan mass spectrum (mass range: *m/z* 100–1500) were subjected to tandem mass spectrometry (MS/MS) analysis: an MS scan time of 0.25 s over 14 min; an MS/MS scan time of 0.06 s; and a 3-step normalized collision energy of 25, 35 and 55. The MS/MS data files were converted from *.RAW* into *.mzXML* files using the MSConvert software and uploaded to the Global Natural Product Social Molecular Networking (GNPS) server, and the molecular networking was performed using the GNPS data analysis workflow employing a special spectral clustering algorithm.

### 3.8. LC-MS of Selected VLC-Derived Marine Cyanobacterial Fractions

Selected VLC-derived marine cyanobacterial fractions were analyzed in the positive-ion-mode electrospray ionization using an MS system (Bruker MAXIS II equipped with a Quadrupole-Time-of-Flight mass analyzer) coupled to an HPLC (Agilent 1290 Infinity equipped with a diode array detector) on a Phenomenex analytical C18 column (2.5 µm, 100 Å, 4.6 × 150 mm). Each fraction was eluted with a starting mobile phase of 5% ACN:95% H_2_O (0.1% formic acid), followed by a gradient of up to 100% ACN for 15 min at a flow rate of 1 mL/min. The raw data file was analyzed using compact Data Analyst (Bruker software) to provide accurate and high-resolution mass per charge molecular ions in the fractions and generate molecular formulae using a Bruker SmartFormula manually.

### 3.9. Molecular Docking of Marine Cyanobacterial Metabolites

The molecular docking of compounds with LasR proteins comprised the following procedures: ligand preparation, protein selection, docking and analysis of the results. Docking was performed with the SwissDock DockingWeb Service (available online: http://www.swissdock.ch/ (accessed on 28 February 2023)). The three-dimensional structures of the autoinducer, *N*-3-oxododecanoyl-L-homoserine lactone and cyanobacterial compounds were either obtained from the PubChem database or created using Chem3D and converted into *.mol2* files using the OpenBabel platform (http://openbabel.org/wiki/Main_Page/ (accessed on 28 February 2022)). The LasR protein structure was retrieved from the Protein Data Bank (PDB) with reference ID 2UV0. The target + ligand set was considered stable when the values of the binding free energy were lower than −7 kcal/mol. This consideration is based on docking experiments with the known X-ray structure (2UV0) complex of the autoinducer and cyanobacterial compounds resulting in binding-energy values ranging from −10.64 to −7.87 kcal/mol. Once the results of the docking were obtained, they were analyzed with UCSF Chimera.

## 4. Conclusions

A new cyclopropane-containing specialized metabolite, benderadiene (**2**) and the known lipid acid lyngbyoic acid (**1**) were purified from samples of the marine cyanobacterium cf. *Lyngbya* sp., collected during bloom season in October 2019. The occurrence of marine cyanobacterial cyclopropyl compounds is not common, and apart from benderadiene, there are at least 13 other cyclopropane-containing molecules reported from marine microalgae. Both lyngbyoic acid and benderadiene showed quorum-sensing inhibitory activity by suppressing fluorescence expression when incubated with the *P. aeruginosa* PAO1 *lasB-gfp* biosensor strain, with the former molecule being more active. In addition, lyngbyoic acid showed anti-biofilm activity when incubated with *Pseudomonas aeruginosa*. A significant reduction in biovolumes was observed when they were treated with concentrations above 500 µM of **1**. The high production of lyngbyoic acid and other free acid forms by marine cyanobacteria, observed in this study and others, provides insights into their multiple ecological roles as defensive molecules and base structures for the synthesis of other structurally related allelochemicals. Free acid forms could provide a chemical defense to marine cyanobacteria via their interference with the bacterial quorum-sensing systems of competing microbes in the environment during bloom seasons. This observation is supported by molecular docking results where free acid forms were found docking at the ligand-binding site of the LasR transcriptional protein.

## Figures and Tables

**Figure 1 molecules-28-03965-f001:**
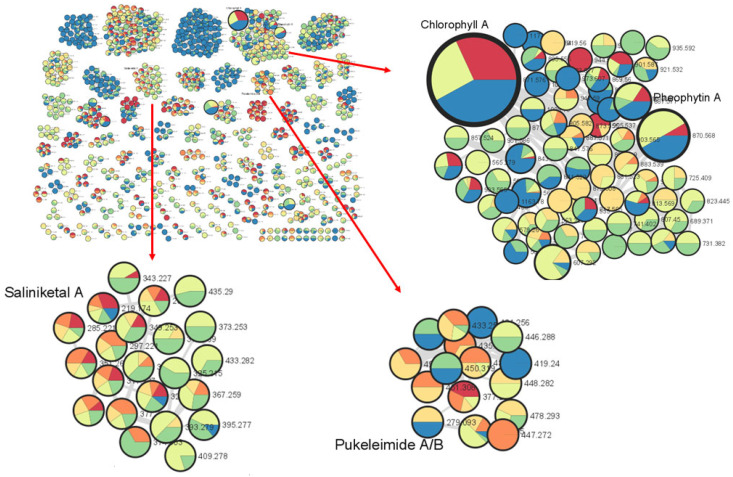
Marine cyanobacterial molecular network with a cosine similarity score cutoff of 0.70. The network was generated when s preliminary extract of cyanobacteria (1 L) was separated into eight fractions based on polarity and then analyzed via LC-MS/MS. The inlaid portions of the network were rearranged in Cytoscape for easier visualization of node connectivity. Compound dereplication based on the GNPS mass spectral database identified clusters of molecular families related to chlorophyll a, pheophytin A, saliniketal a and pukeleimides A/B. The red arrows point to enlargements of selected clusters.

**Figure 2 molecules-28-03965-f002:**
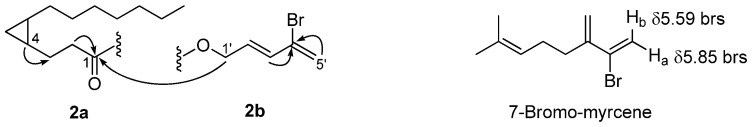
Partial structures of **2a** and **2b** with arrows indicating ^1^H-^13^C HMBC correlations and 7-bromo-myrcene, where the two protons attached to the terminal sp^2^ carbon atom were found to resonate at δ_Ha_ 5.85 and δ_Hb_ 5.59.

**Figure 3 molecules-28-03965-f003:**
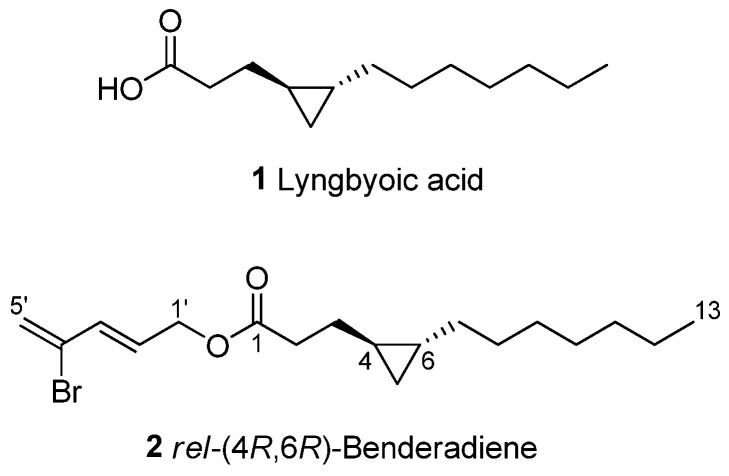
Chemical structures of lyngbyoic acid (**1**) and benderadiene (**2**).

**Figure 4 molecules-28-03965-f004:**
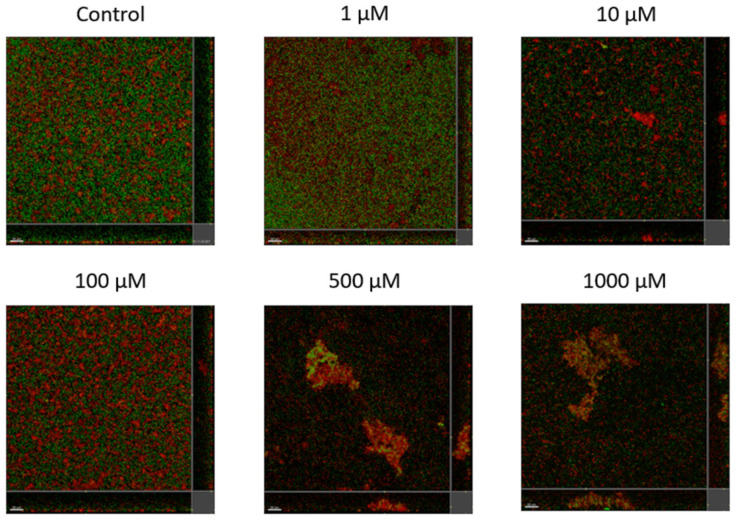
Confocal images of gfp-tagged *P. aeruginosa* biofilms (green) and dead bacteria or extracellular DNA stained with propidium iodide (red) grown in the presence of different concentrations of lyngbyoic acid (**1**).

**Figure 5 molecules-28-03965-f005:**
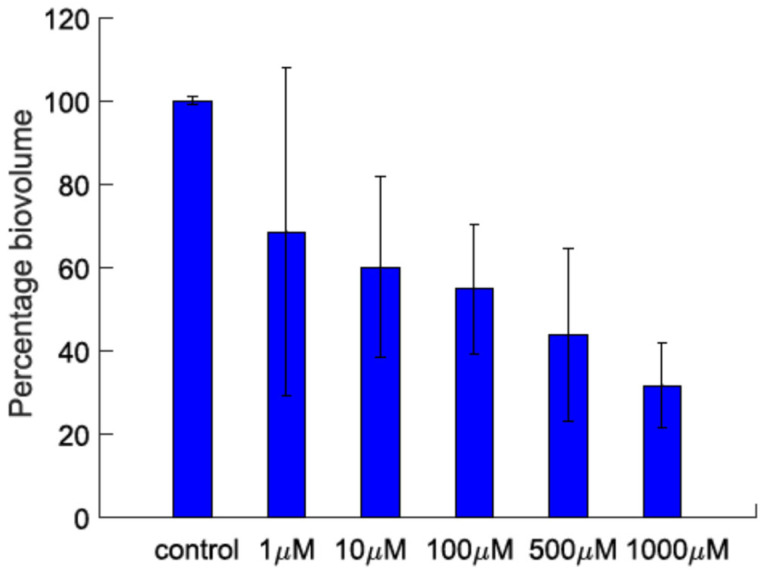
The biovolume percentage of *P. aeruginosa* biofilms grown in the presence of different concentrations of lyngbyoic acid (**1**).

**Figure 6 molecules-28-03965-f006:**
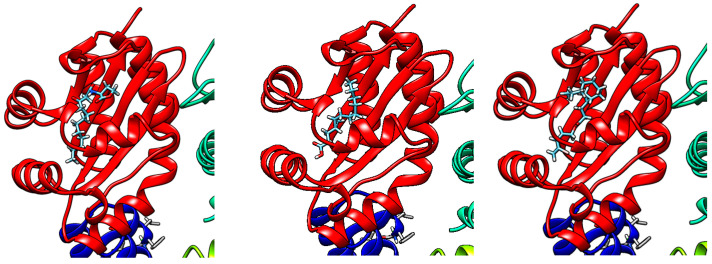
Molecular docking of *N*-3-oxo-dodecanoyl-L-homoserine lactone (**left**), **1** (**middle**) and **2** (**right**), onto the LasR-ligand binding domain using Swissdock and visualized with UCSF Chimera.

**Figure 7 molecules-28-03965-f007:**
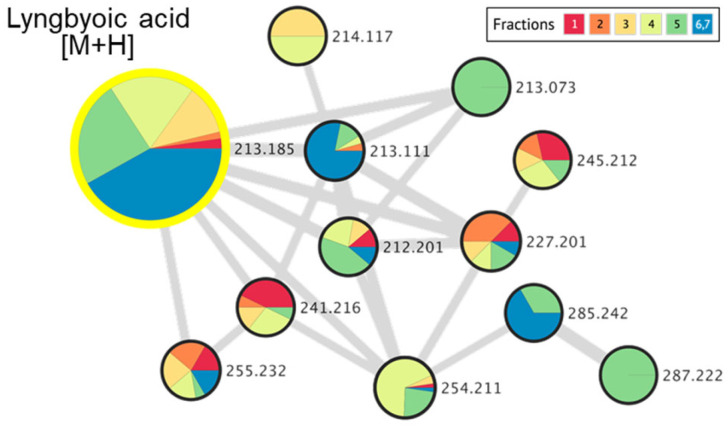
Molecular networking cluster, indicating a molecular family related to lyngbyoic acid and related analogs, detected in marine cyanobacterial VLC-derived fractions 1 to 7.

**Figure 8 molecules-28-03965-f008:**
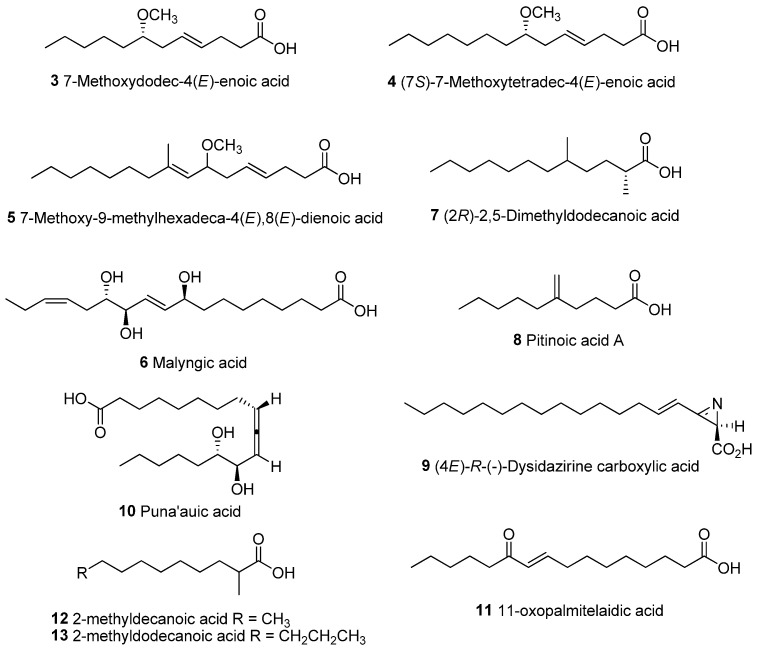
Chemical structures of unique lipid acids isolated from marine cyanobacteria.

**Figure 9 molecules-28-03965-f009:**
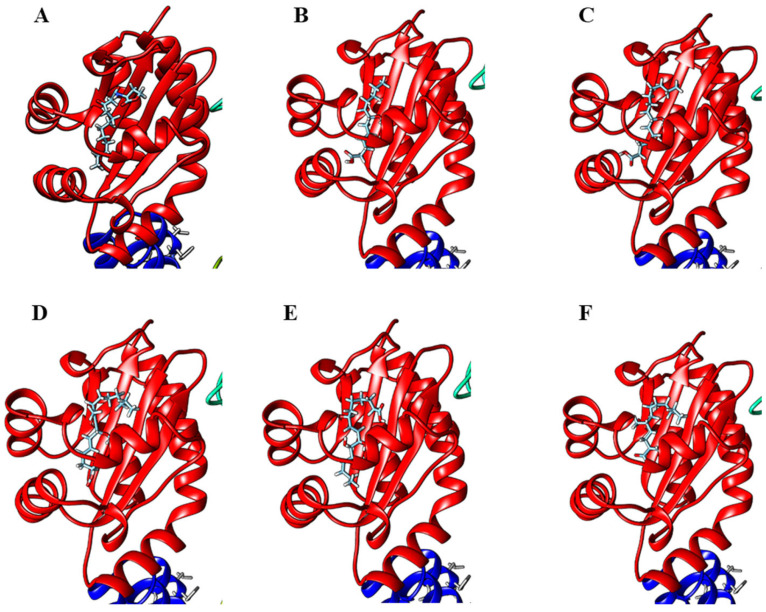
Molecular docking of (**A**) natural ligand, *N*-3-oxo-dodecanoyl-L-homoserine lactone, (**B**) 7-methoxydodec-4(*E*)-enoic acid (**3**), (**C**) 7(*S*)-methoxytetradec-4(*E*)-enoic acid (**4**), (**D**) 7-methoxy-9-methylhexadeca-4(*E*),8(*E*)-dienoic acid (**5**), (**E**) malyngic acid (**6**) and (**F**) (2*R*)-2,5-dimethyldodecanoic acid (**7**) onto the LasR-ligand-binding domain performed on Swissdock and visualized with UCSF Chimera.

**Table 1 molecules-28-03965-t001:** One-dimensional and two-dimensional NMR spectral data of benderadiene (**2**) measured in CDCl_3_.

C Atom	*δ_H_* (m, *J* in Hz)	*δ_C_*	HMBC
1′	4.70 (d, 5.4)	63.0	131.2, 130.6, 173.2
2′	6.16 (dt, 15.8, 11.6)	130.6	128.7, 131.2, 63.0
3′	6.27 (d, 16.0)	131.1	128.7, 130.6, 63.0
4′		128.5	
5′	5.84 (brs)	120.8	128.7
	5.67 (brs)		128.7
1		173.1	63.0, 34.2
2	2.42 (t, 16.7)	34.2	173.2, 29.4, 18.7
3	1.54 (m)	29.4	34.2
4	0.43 (m)	18.7	34.2, 29.4, 11.6
5	0.20 (m)	11.6	34.2, 29.4
6	0.43 (m)	17.9	
7	1.22 (m)	33.8	29.2
8	1.26 (m)	29.2	
9–11	1.25 (m)	31.6	
12	1.28 (m)	22.4	
13	0.87 (t, 13.7)	13.9	

## Data Availability

Not applicable.

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
