# Peer review of "Cyclopropane-Containing Specialized Metabolites from the Marine Cyanobacterium cf. *Lyngbya* sp."

_molecules, 2023, doi:10.3390/molecules28093965_

Round 1
Reviewer 1 Report
This review concerns the article-type manuscript entitled “Cyclopropane-Containing Specialized Metabolites from the Marine Cyanobacterium cf. Lyngbya sp.” and submitted to Molecules journal (Manuscript ID molecules-2359379).
The submitted work concerns isolation, characterization, and evaluation of biological activity of benderadiene (2), an ester derivative of lyngbyoic acid (1), where the alkoxy residue is constituted by 4-bromo-(2E)-2,4-pentadien-1-ol. The area of my expertise is structure determination and the review is focused on this part of the manuscript.
In my opinion the major revision is required.
The following remarks should be considered in the revised version.
A. The supplementary does not help in confirmation of the structure of benderadiene (2).
1. Please, place the HR-QTOF MS spectrum, as it is shown in the section 3.4. Compound Characterization Data.
2. Please, improve Figure S3. 1H NMR spectrum of 2 in CDCl3. The figure should have good quality and contains positions of the signals (only those assigned to the structure (2)), integration of the signals, zoom of the selected range when there is an overlap of the signals, and the structure of benderadiene (2) with numeration, which will clearly indicate which signal belongs to which protons.
3. Please, improve Figure S4. 13C NMR spectrum of 2 in CDCl3. The figure should have good quality and contains positions of the signals (only those assigned to the structure (2)). The structure of benderadiene (2) with numeration, which will clearly indicate which signal belongs to which carbin should be placed for convenience of readers. Additional NMR experiment, for example, DEPT spectrum should be recorded and presented. The benderadiene (2) is a relatively simple molecule, thus, the most information must be retrieved from 1D spectra (1H and 13C) and this information must be clearly presented.
4. Analogously, the 2D spectra presented in Figures 5-7S should be much better presented, and the most important information should be clearly shown.
5. In the main text (line 218) there is information about NOESY experiment, but this spectrum is not presented in the Supplementary.
6. The stereochemistry of cyclopropane ring should be also proven in a better way. The reference [17] (Kwan et al.) is given, but there is only confirmation of the relative configuration trans of the cyclopropane ring. The absolute configuration (4R,6R) is only proposed, based on another reference. This is not enough. Please, take an effort to establish the absolute configuration. If not, the prefix “rel” should be applied, which indicates that it is only a relative configuration, according to IUPAC nomenclature.
B. The structure of benderadiene (2) is relatively small. Thus, an effort to give full IUPAC name should be undertaken, not only trival name should be given. Ii would be nice to have also SMILE and/or InChI code, for convenience of reader and future data gathering using AI methods.
C. The UV spectra should be also valuable in Supplementary, as it is shown in the section 3.4. Compound Characterization Data.
Kind Regards
Reviewer 2 Report
This manuscript reports a new cyclopropane-bearing metabolite, named benderadiene, along with previously-reported lyngbyoic acid (1) from a marine cyanobacterium Lyngby sp. The structure of the new compound was confidently determined mainly by NMR and MS spectroscopic analysis. Benderadiene (2) is indeed a new compound bearing a cyclopropane and a bromine atom. Besides the structure elucidation, the biological activity of the cyclopropane-containing metabolites was evaluated by the quorum sensing-controlled green fluorescent protein (GFP) expression assay. In the assay, compounds 1 and 2 inhibited quorum sensing-regulated gene expression in a dose-dependent manner. Benderadiene (2) showed lower activity than 1. Moreover, molecular docking of the compounds to LasR, a transcriptional activator of Pseudomonas aeruginosa, enabled to propose the molecular level-based mechanism of the compounds showing the inhibitory activity in quorum sensing-controlled GFP expression. The significance is extended by analyzing the ecological function of cyanobacterial fatty acids.
This is a very comprehensive manuscript regarding cyclopropane-bearing cyanobacterial metabolites including structure elucidation, biological evaluation, and ecological significance. Therefore, the reviewer believes that this manuscript will be a very nice reference for follow-up studies about cyclopropane-containing cyanobacterial compounds in the future. In this sense, the reviewer strongly suggests that this manuscript should be published in Molecules after minor revision.
It is well written and there is little to revise. Please consider the following points for minor revision.
(1) Because compound 2 is new, it would be better to have IR data. Lamda and epsilon should be Greek letters in the physicochemical data in 3.4.
(2) Please provide the dimension of the open column and Sep-Pak cartridge.
(3) The coupling constants of the protons at 1’ and 2’ look strange because they are mutually coupled but they do not have a common coupling constant.
(4) Some references have issue numbers but some do not. Please check the reference format.
Reviewer 3 Report
This article describes the molecular structures (1 and 2) and their QS inhibitory effect from Lygnbya sp. along with ecological roles of cyanobaterial fatty acids.
The effect of cyclopropane-containing fatty compounds on QS inhibitory effect through a molecular docking tool was well depicted and furthermore these compounds and other modified fatty acids showed the relationship with cyanobacterial blooms. However, this article has big points to be explained.
1. Supplementary file does not exist in this review version.
2. One of key points in this article is compound 2 (benderadiene). Because of small amount of 2, the purity is important, but I can't see the 1H NMR. So the supplementary file is very needed. in Line 221, a optical rotaton of 2 is similar to that of lyngbyotic acid in Line 221, but in fact, two values are different (2 is +484.7 and lyngbyoic acid is -15.5). This means a different configuration of 2. Explain this.
3. Authors used the bloomed samples. Indicate cell counts (cells/ml).
And minor points are listed below
1. Line 60 : are containg cyclopropane
2. Line 83, not b-oxidation, but beta-oxidation
3. Line 174, C18H29BrO2
4. Line 174, ~ requiring four degrees of unsaturation.
5. Line 219, "trans-" to italic style
6. Line 231, P. aeruginosa to Pseudomonas aeruginosa
7. Line 237, Pseudomonas aeruginosa to P. aeruginosa
8. Line 268, ~ Cyanobacteroal Cyclopropane-". delete "of"
9. Line 413, compounds 1 and 2,
10. VFC in the text ( Line 475-502) to VLC
Good
Round 2
Reviewer 1 Report
The review concerns the revised version of the article-type manuscript entitled “Cyclopropane-Containing Specialized Metabolites from the Marine Cyanobacterium cf. Lyngbya sp.” and submitted to Molecules journal (Manuscript ID molecules-2359379).
My previous remarks related to the determination of the structure of benderadiene (2). In general, Authors respond on my questions and improved the article in this part:
1. Mass spectra were placed as Figure S2.
2. SMILE notation and InChI code for benderadiene (2) was presented in Table S2.
3. UV and HR-QTOF MS spectra of 2 were presented as Figure S3.
4. Figure S4. 1H NMR spectrum of 2 in CDCl3 was considerably improved. Positions of the signals are given as well as the integrations of separated signals. The structure is given and the signals are assigned to the protons.
5. Figure S5. 13C NMR spectrum of 2 in CDCl3 is improved. The signals are assigned to the carbons.
6. Figure S6. DEPT spectra of 2 in CDCl3 is presented.
7. The stereochemistry (E for double bond and trans in relation to cyclopropane) is shown, although in case of cyclopropane the relative stereochemistry is determined.
In summary, the work can be accepted.
Remarks:
1. Figure S5. Please assign signal at 60.02 ppm to C1’.